# Identifying the key regulators that promote cell-cycle activity in the hearts of early neonatal pigs after myocardial injury

Eric Zhang[1], Thanh Nguyen[2], Meng Zhao[1], Son Do Hai Dang[3], Jake Y. Chen[2], Weihua Bian[1], Gregory P. Walcott[1,4]*

1 Department of Biomedical Engineering, the University of Alabama at Birmingham, Birmingham, AL, United States of America, 2 Informatics Institute, the University of Alabama at Birmingham, Birmingham, AL, United States of America, 3 Department of Computer Science, the University of Alabama at Birmingham, Birmingham, AL, United States of America, 4 Department of Medicine/Cardiovascular Diseases, the University of Alabama at Birmingham, Birmingham, AL, United States of America

* gregorywalcott@uabmc.edu

**Data Availability Statement:** All relevant data are within the paper and its Supporting Information files.

## Abstract

Mammalian cardiomyocytes exit the cell cycle shortly after birth. As a result, an occurrence of coronary occlusion-induced myocardial infarction often results in heart failure, postinfarction LV dilatation, or death, and represents one of the most significant public health morbidities worldwide. Interestingly however, the hearts of neonatal pigs have been shown to regenerate following an acute myocardial infarction (MI) occuring on postnatal day 1 (P1); a recovery period which is accompanied by an increased expression of markers for cell-cycle activity, and suggests that early postnatal myocardial regeneration may be driven in part by the MI-induced proliferation of pre-existing cardiomyocytes. In this study, we identified signaling pathways known to regulate the cell cycle, and determined of these, the pathways persistently upregulated in response to MI injury. We identified five pathways (mitogen associated protein kinase [MAPK], Hippo, cyclic [cAMP], Janus kinase/signal transducers and activators of transcription [JAK-STAT], and Ras) which were comprehensively upregulated in cardiac tissues collected on day 7 (P7) and/or P28 of the P1 injury hearts. Several of the initiating master regulators (e.g., CSF1/CSF1R, TGFB, and NPPA) and terminal effector molecules (e.g., ATF4, FOS, RELA/B, ITGB2, CCND1/2/3, PIM1, RAF1, MTOR, NKF1B) in these pathways were persistently upregulated at day 7 through day 28, suggesting there exists at least some degree of regenerative activity up to 4 weeks following MI at P1. Our observations provide a list of key regulators to be examined in future studies targeting cell-cycle activity as an avenue for myocardial regeneration.

## Introduction

We and others have reported that hearts of neonatal pigs are able to regenerate myocardial tissue lost to an acute myocardial infarction (MI) that occurs within the first two days following birth (postnatal day [P] 1) [1, 2]. On P28, measurements of left-ventricular (LV) anterior wall

**Funding:** At the time of working on this study and the manuscript, all of the authors (EZ, TN, MZ, SDHD, JC, WB and GPW) were employed by GoodAI Research s.r.o. The funder provided support in the form of salaries for all authors, provided the hardware for performing experiments and approved the final decision to publish. The funder did not have any additional role in the study design, data collection and analysis, or preparation of the manuscript. This study was funded in part by the NIH. No additional external funding was received for this study. The specific roles of the authors are articulated in the 'Author Contributions' section.

**Competing interests:** Authors EZ, TN, MZ, SDHD, JC, WB and GPW were employees of GoodAI Research s.r.o. This does not alter our adherence to PLOS ONE policies on sharing data and materials. The company does not hold any patents pertaining to the work described in the paper, and there are no other restrictions on the sharing of data and/or materials published in this manuscript.

thickness reached 90% of measurements of those in age matched control animals and were accompanied by an increase in the expression of markers for cell-cycle activitiy and proliferation (Ki67, PH3, and Aurora B) compared again to age matched control animals. Furthermore, the correlation between regenerative capacity and cell-cycle marker expression suggests that myocardial regeneration in very young animals likely occurs by means of proliferation of endogenous cardiomyocytes. Consequentlyupstream regulators of cell-cycle activity may be effective therapeutic targets for promoting myocardial proliferation in older mammals [3, 4]. Most experimental strategies for enhancing myocardial recovery through modulating signaling pathways have focused on a single gene, growth factor, or other biologically active molecule [5, 6], and a number of studies have targeted the upstream regulators of major signaling cascades such as p38/MAP kinase [7], the Ras family of GTPases [8], or the transcription factors TBX5 [9] and GATA4/6 [10, 11] in older large animals. However, there have been few studies examining the cardiomyocyte proliferative window in large young mammals that correlate gene expression to corresponding protein levels and is coupled with in-vivo analysis [12].

The emerging field of bioinformatics provides new tools that can help scientists identify the regulatory steps influencing myocardial cell cycle and other biological processes. The analysis is performed in two steps. First, sequencing data generated from the experimental and control groups are compared to produce a list of differentially expressed genes (DEG) [13, 14]; then, pattern-matching algorithms [15, 16] compare the DEG list to databases of known pathways to determine which pathways are most consistent with the observed variations in gene expression. This approach has been widely used in computer simulations of cardiac regeneration; however, the databases typically comprise an aggregate of results from experiments performed in different cell and tissue types. As such, the results may not be accurate for a specific tissue of interest (e.g., myocardium). Because such analysis provides only a ranked list of pathways, it cannot be used to identify which components within the pathways may be targeted to validate the predicted outcome in-vivo or to produce a therapeutic effect [17].

For this report, we conducted the two steps of typical pathway analysis in reverse. First, we queried the KEGG Pathway Database to identify signaling cascades known to regulate cell-cycle and cell-fate determination [17]; and then used the results from our gene expression analyses [18] of pig heart samples undergoing MI at P1 and their age-matched normal from our previous study [19], to show that that five of the twenty putative cell-cycle/cell-fate pathways were up-regulated for 4 weeks in response to MI injury at P1. Collectively, our observations provide a comprehensive list of molecular targets for future in-vitro and in-vivo studies of myocardial regeneration.

## Methods

### Animals

All experimental protocols were approved by the Institutional Animal Care and Use Committee of the University of Alabama Birmingham (UAB) and performed in accordance with the National Institutes of Health Guide for the Care and Use of Laboratory Animals (National Institutes of Health publication No. 85–23). Pigs (Prestage Farms, Inc, West Point, MS) were fed every 4 hours with bovine colostrum on P1-P2, with a 1:1 ratio of colostrum:sow's milk on P3, and with cow's milk from P4 until sample collection. Supplemental iron was provided on P7, and the animals were housed in an incubator at ~85˚F with room air until either P7 or P28 days of age. Surgical procedures are detailed in our previous report [19]; briefly, MI was induced via permanent ligation of the left descending coronary artery. Seven and 28 day old normal animals were used as controls.

## RNA sequencing

RNA sequencing was performed according to the Illumina NGS protocol [20]. RNA was isolated from tissue samples (~30 mg) collected from the border zone of ischemia in the hearts of MI animals and from the corresponding region of hearts in age-matched normal animals. Libraries were quantified via qPCR with equal amounts of RNA for all samples, and RNA concentrations were diluted to 3 nM before loading equimolar amounts onto the flow cells.

Sequencing data were processed in the Cheaha computer cluster at UAB (https://docs. uabgrid.uab.edu/wiki/cheaha). Paired-end fastq files were evaluated with the trim-galore [20] package for quality assessments of the input sequencing files, and mapping was conducted with the Ensembl pig genome assembly (Sscrofa11.1) [21] and STAR package v2.5.2 [22] to produce the sorted BAM (Binary Alignment Map) file format; following this step, the sorted BAM file was converted into the SAM (Sequence Alignment Map) format with SAMtools [23], and mapped transcripts (raw expression) were counted with the HTSeq/0.6.1 package [24]. Expression levels were normalized and differential gene expressions (DEG) ($\log_2$ of the fold-change and p-value) were analyzed with Deseq2 software [25]. In 3x3 comparsions, the probability of observing as extreme and more than 'all A-samples strictly greater than (or less than) B-samples' would be $0.5^9 = 0.002$. This is the p-value for the statistical result above. The p-values other cases are computed as in [23].

## Pathway analysis

Focusing on cell cycle regulators, we used DEG results from MI and age-matched control heart samples to annotate the following KEGG[26] swine pathways:

1. MAPK signaling pathway: The MAPK signaling pathway participates in many cellular functions, including cell proliferation, differentiation, and migration [27]. This pathway is critical in bridging from growth factor signaling to cell cycle intiation in mammalian cells [28].

2. HIPPO signaling pathway: In this pathway, LATS1/2 phosphorylates the transcriptional coactivators YAP and TAZ and leads to apoptosis at high cell density [29]. Previous studies show that a deficiency of HIPPO signaling pathway is able to reverse systolic heart failure following myocardial infarction [30].

3. RAS signaling pathway: This pathway functions as molecular switches in regulating cell proliferation, survival, growth, migration, differentiation and cytoskeletal dynamics [31].

4. VEGF signaling pathway: VEGF activates Protein Kinase C gene expression [32]; PKC enters the RAS signaling pathway. There has been evidence showing the correlation between VEGF expression and reduce cardiomyocyte apoptosis [33].

5. WNT signaling pathway: Wnt genes and their receptors stabilize beta-catenin. Beta-catenin activates cell-cycle transcription factors, such as LEF1 and MYC [34]. The HIPPO pathway could inhibit WNT signaling pathway and restrain cardiomyocyte proliferation [35].

6. RAP1 signaling pathway: This pathway controls the cell division process [36].

7. Hedgehog signaling pathway. This pathway regulates differentiation, proliferation, tissue polarity, stem cell population and carcinogenesis [37].

8. JAK-STAT signaling pathway: The JAK/STAT pathway is the principal signaling mechanism for many cytokines and growth factors [38].

9. cAMP signaling pathway: cAMP acts directly on protein kinase A and Epac genes [39, 40]. These genes activate the RAP1 and MAPK pathways, which regulates calcium homeostasis, muscle contraction, cell fate, and gene transcription [41].

10. ERBB signaling pathway: ERBB genes bind extracellular growth factor ligands to the MAPK signaling pathway [42], which regulates proliferation and differentiation.

11. cGMP-PGK signaling pathway [43, 44]: In cardiac myocyte, PKG directly phosphorylates a member of the transient potential receptor canonical channel family, TRPC6, suppressing this nonselective ion channel's Ca2+ conductance, G-alpha-q agonist-induced NFAT activation, and myocyte hypertrophic responses.

12. Apelin signaling pathway: This pathway regulates angiogenesis, cardiovascular functions, cell proliferation [45, 46].

13. NF-kappa B signaling pathway: This pathway controls transcription of DNA, cytokine production and cell survival [47]

14. TNF signaling pathway: This pathway regulates the immune cells, which are recruited as response to injury. The pathway could induce apoptosis [48].

15. HIF-1 signaling pathway: implicated in proliferation of fetal cardiomyocytes [49].

16. FoxO signaling pathway. FoxO signaling pathway includes forkhead-box transcription factors regulating cell-cycle control [50].

17. Sphingolipid signaling pathway: implicated in satellite muscle cell cell-cycle initiation [51].

18. Phospholipase D signaling pathway. Phospolipase D signaling pathway regulates the production of the lipid second messenger phosphatidic acid. Posphatidic acid is involved in many biological processes, including cell-cycle activity [52].

19. mTOR signaling pathway. mTOR signaling pathway serves as a central regulator of cell metabolism, growth, proliferation and survival [53].

20. AMPK signaling pathway. AMPK signaling pathway regulates ATP production and consumption; therefore, it has a critical role in regulating growth [54].

For each gene in the above pathways, we determined which genes met the following criteria:

i. Pro-proliferative and cell-cycle genes expressed at a significantly higher level in MI samples at P7 compared to controls. Average expression in MI-P7 samples are *at least two-fold higher* than CTL-P7 samples. MI-P7 error bars were required to be *strictly higher* than CTL-P7 error bars.

ii. Of genes which met criteria in i.), also had at least *two-fold* differential expression when comparing MI vs CTL at the sample collection day P28

**Western blotting.** Tissues were lysed in Mammalian Protein Extraction Reagent (Fisher scientific, PI78501) with protease (Sigma, 04693116001) and phosphatase (Sigma, P0044) inhibitors; then, the lysates were denatured at 100˚C for 6 min, separated in a 4–20% precast gel (Bio-rad, 4568093), and transferred onto a PVDF membrane (Bio-rad, 1704156). The membrane was incubated with 5% non-fat milk (Bio-rad, 1706404) for 30 min, with primary antibodies at 4˚C overnight, and then with horseradish-peroxidase (HRP)–conjugated secondary antibodies for 30 min. Protein bands were detected with the chemiluminescent HRP substrate (Millipore, WBKLS0500) in a ChemiDocTM Imaging System (Bio-rad). Phosphorylated proteins were detected first; then,

the blots were treated with stripping buffer (Fisher scientific, PI21059) before total protein expression was evaluated.

## Results

Pigs underwent induction of MI one day after birth (MI-P1), and gene-expression levels of pathway components in border-zone infarcted heart samples were compared to corresponding regions from hearts of age-matched controls, at P1, P7, and P28, via bulk RNA sequencing. Early and late postnatal genes were defined as those found to be upregulated in MI-P1 animals at P7 and P28, respectively. A minimum of 200 transcripts (raw count) was required for each gene from all samples, and upregulation was defined as an expression level that was at least two-fold greater in MI tissues compared to age-matched control tissues. For early-postnatal genes, upregulation was also required to have no increase in expression from P1 to P7 in control samples, alongside no overlap between error bars in P7-MI and P-7 controls. Of the 11 general signaling pathways that were associated with the cell cycle and cell fate in the KEGG Pathway Analysis database, five were comprehensively upregulated, from initial signaling molecule (e.g., growth factors and receptors) to final effector molecules (e.g., transcription factors and proliferative genes) at P7 and/or P28. The results for these five fully upregulated pathways are described in more detail below.

### Mitogen-Activated Protein Kinase (MAPK)

MAPK signaling participates in many cellular functions, including proliferation, differentiation, and migration [27], and serves as a crucial link between extracellular growth factors and the regulation of cell cycle proteins in mammalian cells [28]. Three subpathways of MAPK signaling (Fig 1), were comprehensively upregulated in hearts of MI-P1 pigs—1.) the canonical MAPK pathway, shown previously to contribute to myocardial regeneration [55] and initiating through activation of the CSF1-CSF1R axis [56], GRB2, SOS1/2, Ras family members KRAS/NRAS, ARAF/RAF1, MAP2K1/2 [57], MAPK1/3 [58], ATF4/ELK4/MYC [59], and SRF/FOS [6]; 2) a second CSF1-intiated pathway that followed the classic MAPK pathway through Ras-family activation before diverging to upregulate MAP3K1 [60], MAP2K4, MAPK8/9/10, FOS, and JUND [61]; and 3) a pathway that was initiated by CD14 and proceeds through upregulation of TAB1/2, MAP3K7 [62], IKBKG [63], NFKB1/2, and RELA/RELB [64]. Both the initial (CSF1, CSF1R, CD14) and terminal (ATF4, FOS, JUND, NFKB1/2, RELA/RELB) components of all three MAPK subpathways continued to be more highly expressed in MI-P1 than in age-matched control samples at P28, suggesting that MI-induced MAPK activation and myocardial regeneration persists for at least 4 weeks after myocardial injury; however, most (but not all) intermediate pathway components were upregulated only during the early-postnatal period (e.g. _________). NF1 and members of the DUSP gene family were also identified among the upregulated early postnatal genes, and function as MAPK inhibitors, suggesting MAPK-mediated myocardial regeneration may be promoted via NF1 and/or DUSP inhibition. Notably, the selective inhibition of DUSP genes has been shown to promote cardiac development [65], while deletion of NF1 in neonatal mice leads to cardiac hypertrophy and premature mortality [66].

### Hippo

The Hippo signaling pathway is controlled by the upstream regulator Moesin-Ezrin-Radixin-Like Protein (MERLIN, encoded by the NF2 gene), and proceeds through Salvador-1 (encoded by SAV1) to activate the LATS1/2-mediated phosphorylation of two transcription factors—YAP and WWTR1. Phosphorylated YAP/WWTR1 is sequestered in the cytoplasm where it is

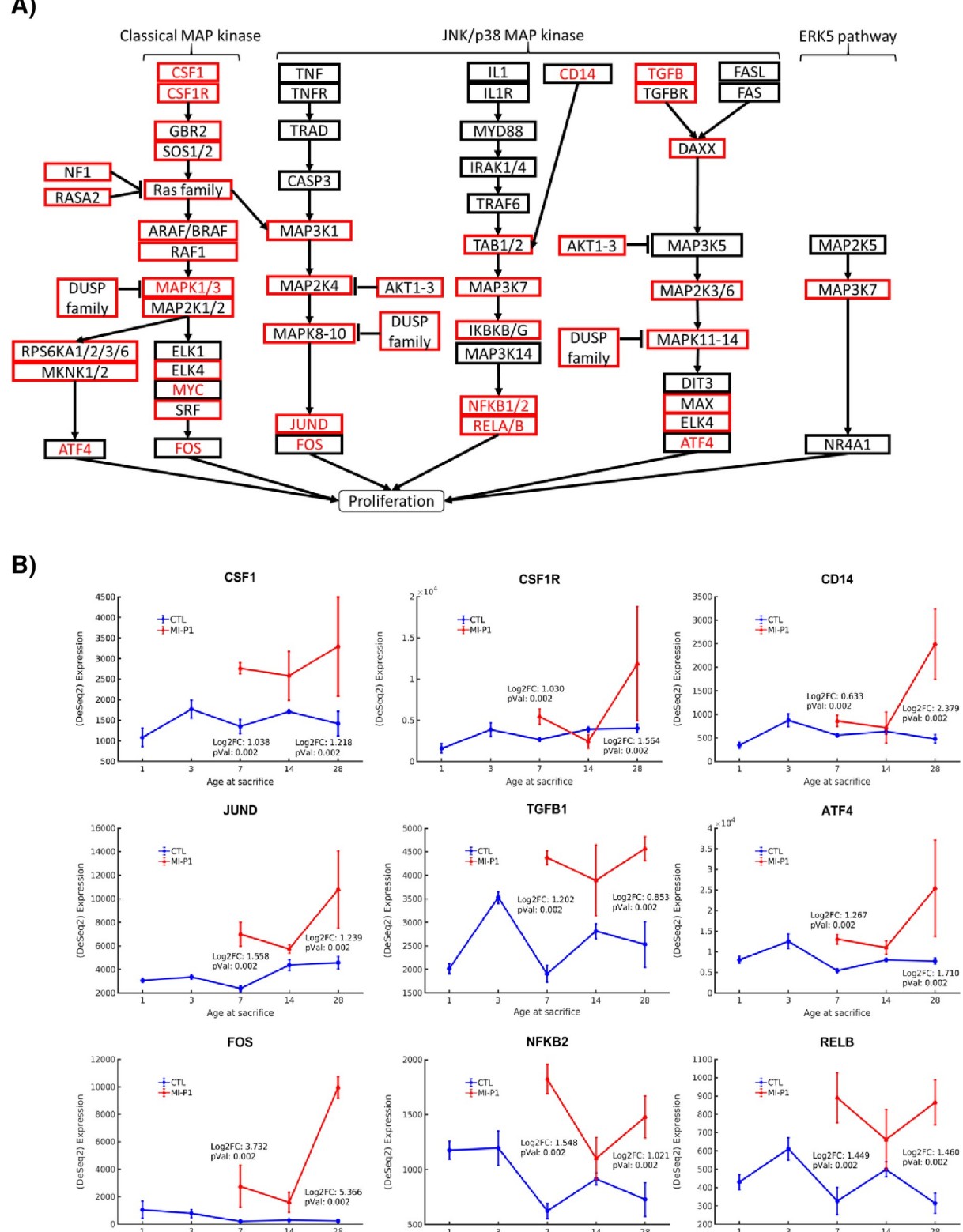

**Fig 1. MAPK signaling is fully upregulated by MI injury in the hearts of 1-day-old pigs.** (A) The subpathways of MAPK signaling are displayed in a flow chart; genes that were expressed at significantly higher levels in the hearts of MI-P1 animals than in Age-matched normal -P1 hearts at P7 (i.e., early postnatal genes) or P28 (i.e., late postnatal genes) are displayed in a red box or in red text, respectively. (B) The

expression of the initial signaling molecules (CSF1/CSF1R, CD14, TGFB) and the terminal effector molecules (JUND, ATF4, FOS, NFKB2 and RELB) of the MAPK signaling pathway was evaluated at the indicted time points in MI-P1 and Age-matched normal -P1 (CTL) hearts.

targeted for degradation, a process which has been shown to promote apoptosis in cultures of confluent cells [29]; however, when Hippo signaling is inhibited, unphosphorylated YAP/ WWTR1 translocates into the nucleus, where it promotes the expression of genes that impede apoptosis and promote proliferation. The Hippo pathway may also suppress cardiomyocyte proliferation via the inhibition of Wnt signaling [35], and previous reports indicate that the inhibition of Hippo signaling may reverse the progression of heart failure in mice [30], while SAV1 and LATS1/2 inhibition can improve myocardial recovery [12]. Our analyses indicate that Hippo pathway components from NF2 through LATS1/2 were upregulated in MI-P1 pigs during the early postnatal period (Fig 2); however, the expression of YAP1/WWTR1 in MI-P1 and their age-matched controls were similar at Day 7 and Day 28, suggesting that the increase in LATS1/2 expression did not impede Hippo signaling via YAP1/WWTR1 degradation. Furthermore, the TGFB-SMAD subpathway of Hippo signaling was fully upregulated in MI-P1 pigs during the early postnatal period, culminating with increases in expression of the proliferative genes FGF1 [67] and ITGB2, and both the initial (TGFB) and terminal (ITGB2) components of the TGFB-SMAD subpathway remained upregulated in MI-P1 pigs on Day 28. Thus, any potential increase in LATS1/2-induced YAP1/WWTR1 phosphorylation and sequestration could have been offset by an increase in TGFB subpathway activity, particularly at later timepoints [68, 69].

## Cyclic AMP (cAMP)

cAMP levels increase in response to binding between G-protein coupled receptors (GPCRs) and their extracellular signaling ligands, and the resulting cAMP signaling cascade activates, notably, protein kinase A (PKA) via adenylyl cyclase, among several other kinases [39, 40], to activate RAP1 and MAPK pathways that regulate calcium homeostasis, muscle contraction, cell fate, and gene transcription [41]. A single initiating ligand, atrial natriuretic peptide (encoded by NPPA) [70], and its corresponding GPCR receptor, ADYCYAPR1R1, were upregulated in MI-P1 swine at Day 7, which led to upregulation of GNAS and ADYC5/6 (Fig 3). ADYC5/6 subsequently activated the classic MAPK subpathway (MAP2K1/2, MAPK1/3, JUN, and FOS) via both the upregulation of PRKCA/B/C and sequential upregulation of RAPGEF3/ 4 and RAP1A/B. Activation of RAP1A/B (Ras protein) expression also resulted in afadin (AFDN gene) upregulation, which promotes actin cytoskeleton rearrangement and effects cell-cell adhesion, 2) the VAV2-RAC1-PAK1 subpathway, which also functions in cytoskeletal remodeling, and 3) PI3K-AKT1/2/3, which stimulates proliferation. Notably, the PRKCA/B/C subpathway also proceeds through the PPP1C family of CREB inhibitors [71] which promote cell proliferation, and PPP1CA/B/C were all upregulated at Day 7, while CREB proteins were downregulated on Day 28, in MI-P1 pigs.

## Janus Kinase/Signal Transducers and Activators of Transcription (JAK-STAT)

The JAK/STAT pathway is the principal signaling mechanism for many cytokines and growth factors [38], and previous experiments have shown that early activation of JAK1/STAT3 is required for the proliferative response to cardiac injury in zebrafish [72]. Four JAK-STAT subpathways were fully upregulated in MI-P1 pigs (Fig 4), including the anti-apoptotic branch that proceeds through PIM1 [73], the MYC–CCND1/2/3 branch, which promotes cell-cycle

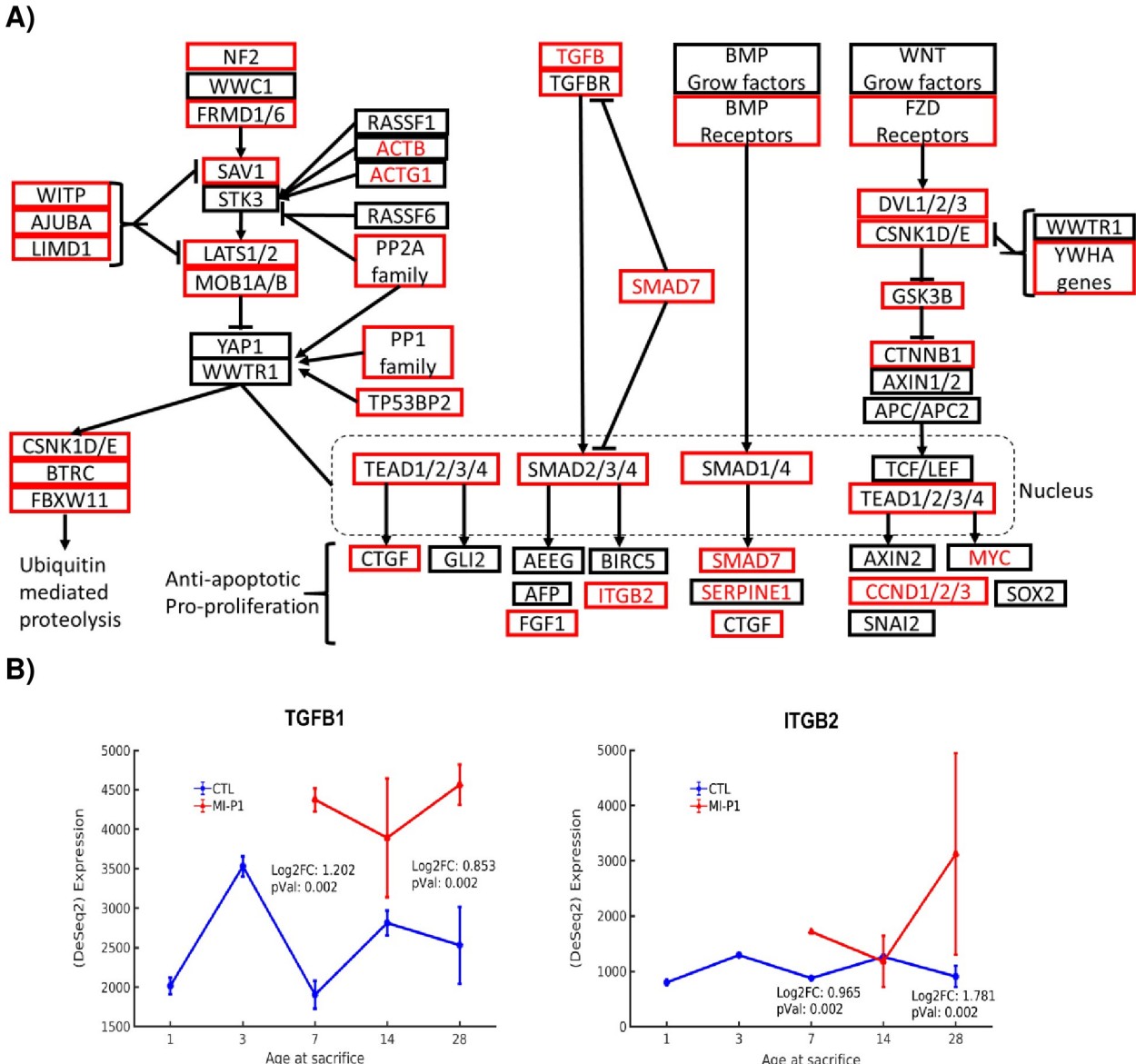

**Fig 2. Hippo signaling is fully upregulated by MI injury in the hearts of 1-day-old pigs.** (A) The subpathways of Hippo signaling are displayed in a flow chart; genes that were expressed at significantly higher levels in the hearts of MI-P1 animals than in Age-matched normal -P1 hearts at P7 (i.e., early postnatal genes) or P28 (i.e., late postnatal genes) are displayed in a red box or in red text, respectively. (B) The expression of an initial signaling molecule (TGFB1) and a terminal effector molecule (ITGB2) of the Hippo signaling pathway was evaluated at the indicted time points in MI-P1 and Age-matched normal -P1 (CTL) hearts.

progression [4, 74], the branch composed of PTPN11/GRB, SOS1/2, HRAS, and RAF1, which regulates proliferation and differentiation, and the branch that proceeds through PI3K, AKT1/2/3 [20], and MTOR to promote the cell cycle and cell survival. The upregulation of PIM1, MYC, and CCND1/2/3 persisted through Day 28.

## RAS

RAS genes function as molecular switches during the regulation of cell proliferation, survival, growth, migration, and differentiation, as well as cytoskeletal dynamics [31]. Our analysis

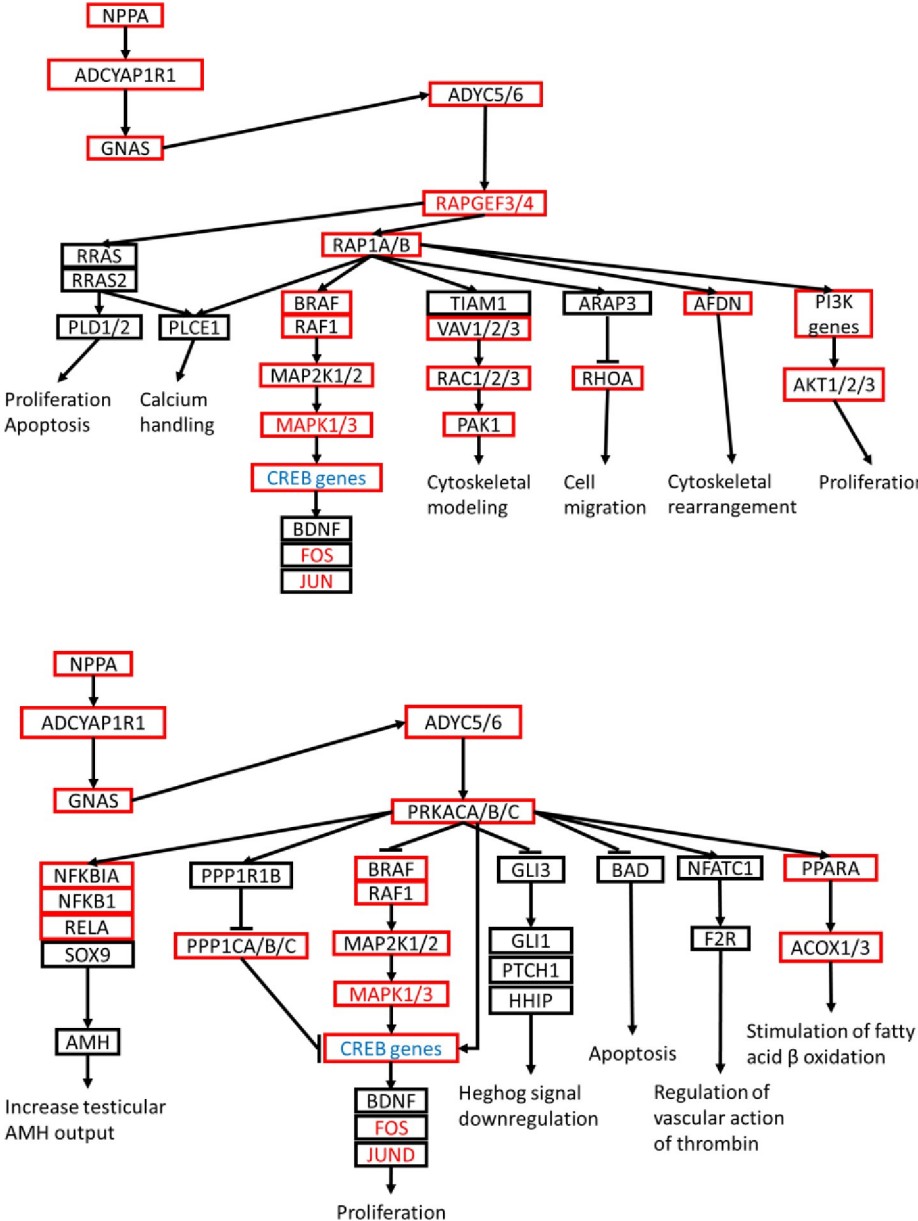

**Fig 3. cAMP signaling is altered by MI injury in the hearts of 1-day-old pigs.** Two subpathways of cAMP signaling are displayed as flow charts. Genes that were expressed at significantly higher levels in the hearts of MI-P1 animals than in Age-matched normal -P1 hearts at P7 (i.e., early postnatal genes) or P28 (i.e., late postnatal genes) are displayed in a red box or in red text, respectively, and genes that were expressed at lower levels in MI-P1 than in Age-matched normal -P1 hearts at P28 are displayed in blue text.

indicated that in addition to MAPK and JAK-STAT signaling, CSF1/CSF1R activation in MI-P1 pigs also fully upregulated subpathways of Ras signaling that progressed through AFDN to control the formation of intercellular junctions, and through PI3K, AKT1/2/3, IKBKG, and NFKB1/RELA to activate cell-cycle progression and promote cellular growth, migration, and survival (Fig 5).

In order to validate the activation of these signaling pathways at protein level, Western blotting experiments were performed and the data are summarized in Fig 6. We have conducted

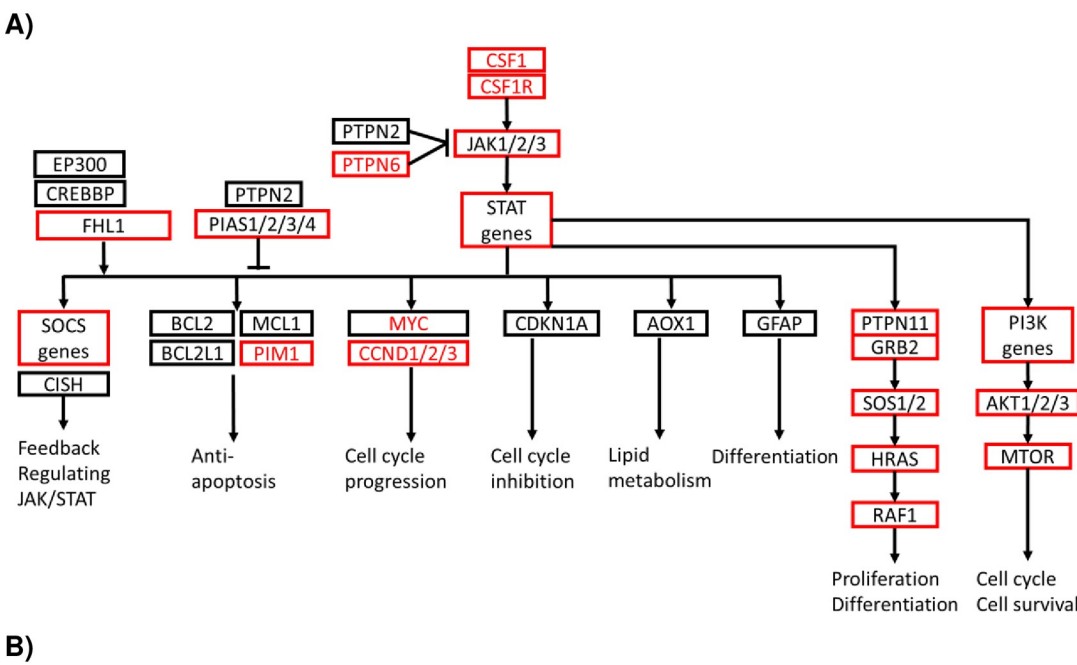

**Fig 4. JAK/STAT signaling is fully upregulated by MI injury in the hearts of 1-day-old pigs.** (A) The subpathways of JAK/STAT signaling are displayed in a flow chart; genes that were expressed at significantly higher levels in the hearts of MI-P1 animals than in Age-matched normal -P1 hearts at P7 (i.e., early postnatal genes) or P28 (i.e., late postnatal genes) are displayed in a red box or in red text, respectively. (B) The expression of three terminal effector molecules (PIM1, MYC, and CCND3) of the JAK/STAT signaling pathway was evaluated at the indicted time points in MI-P1 and Age-matched normal -P1 (CTL) hearts.

Western blot analyses of the expression of MAPK1/3, which mediates MAPK signaling; Akt, which mediates the cell-cycle regulatory activity of the JAK/STAT, cAMP, and RAS pathways; and the Wnt regulators GSK 3α/β and β-catenin, which act on downstream factors of the Hippo pathway. The amount of activated (i.e., phosphorylated) 42/44-MAPK and Akt in MI P1 LAD ligation hearts at page P7 or P28 was significantly greater in MI hearts than in age matched normal (Fig 6); thus, MAPK, Wnt/β-catenin and Akt signaling were upregulated at protein level for at least 28 days after LAD ligation at P1.

## Discussion

Mammalian cardiomyocytes undergo cell cycle arrest in the early post-natal period; however, we have shown that when MI occurs in the hearts of 1-day-old piglets, endogenous

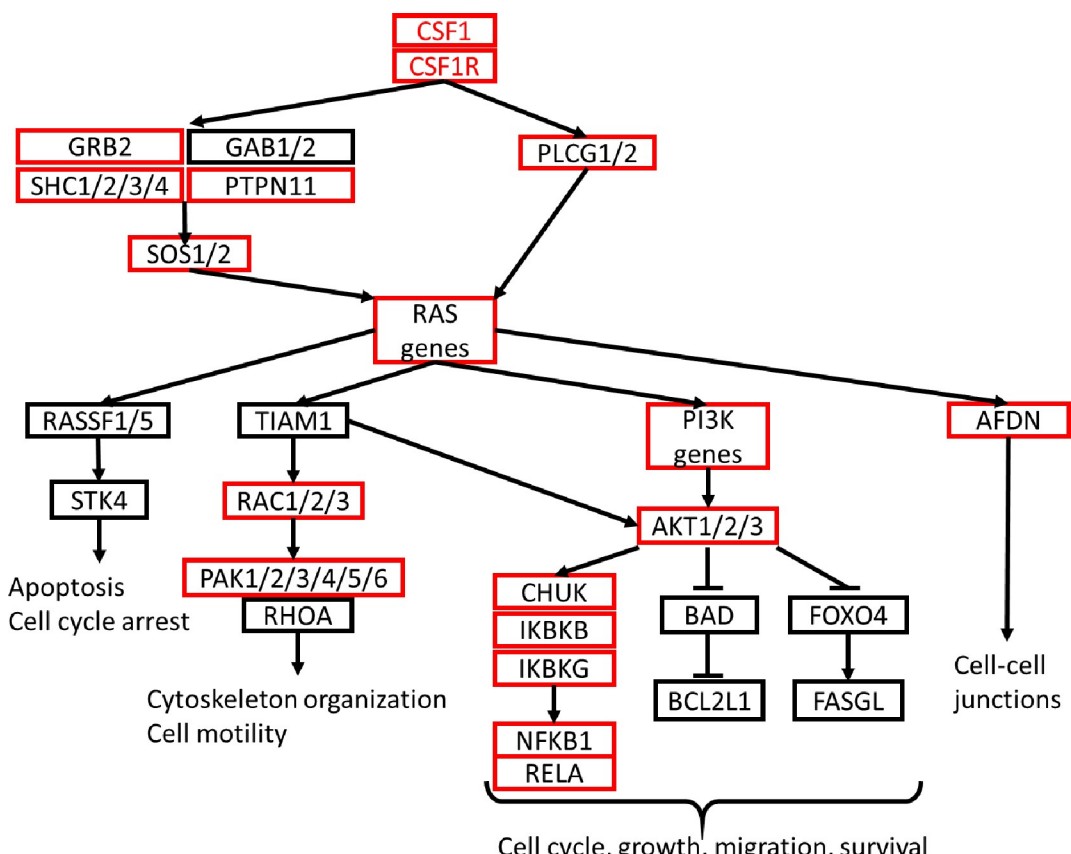

**Fig 5. RAS signaling is fully upregulated by MI injury in the hearts of 1-day-old pigs.** The subpathways of RAS signaling are displayed in a flow chart; genes that were expressed at significantly higher levels in the hearts of MI-P1 animals than in Age-matched normal -P1 hearts at P7 (i.e., early postnatal genes) or P28 (i.e., late postnatal genes) are displayed in a red box or in red text, respectively.

cardiomyocytes proliferate to regenerate the injured myocardium[19]. Here, we present the results that shows which signaling pathways are responsible for MI-induced cardiomyocyte cell-cycle activation and proliferation Data from this study show that 5 of 20 possible regulatory pathways are upregulated in this model of early myocardial infarction in a large animal.

In conventional pathway analysis [75], differences in gene expression between control and experimental groups are used to search a pathway database to determine which pathways may be up- or down-regulated in response to the experimental interventions overall. In our current investigation, the KEGG Pathway Database was queried first, to identify signaling cascades that are known to regulate the cell cycle, cell-fate determination, and proliferation; then, the results from differential gene expression analyses with tissues from the hearts of MI-P1 or age-matched normal animals [19] were used to identify which of the known pathways were activated when MI was surgically induced one day after birth There are two advantages to this strategy. First, by choosing pathways that we are interested in a priori, we can test whether or not they are up regulated using unbiased measures. Second, we learn what pathways are likely not involved, thus giving a fuller picture of the process we are studying.

Five pathways (MAPK, Hippo, cAMP, JAK/STAT, and Ras) were upregulated at all points from the initiating master regulators (e.g., CSF1/CSF1R, TGFB, and NPPA) to the terminal effector molecules (e.g., ATF4, FOS, RELA/B, ITGB2, CCND1/2/3, PIM1, RAF1, MTOR, NKF1B), Although several of the upregulated genes are known to contribute to myocardial

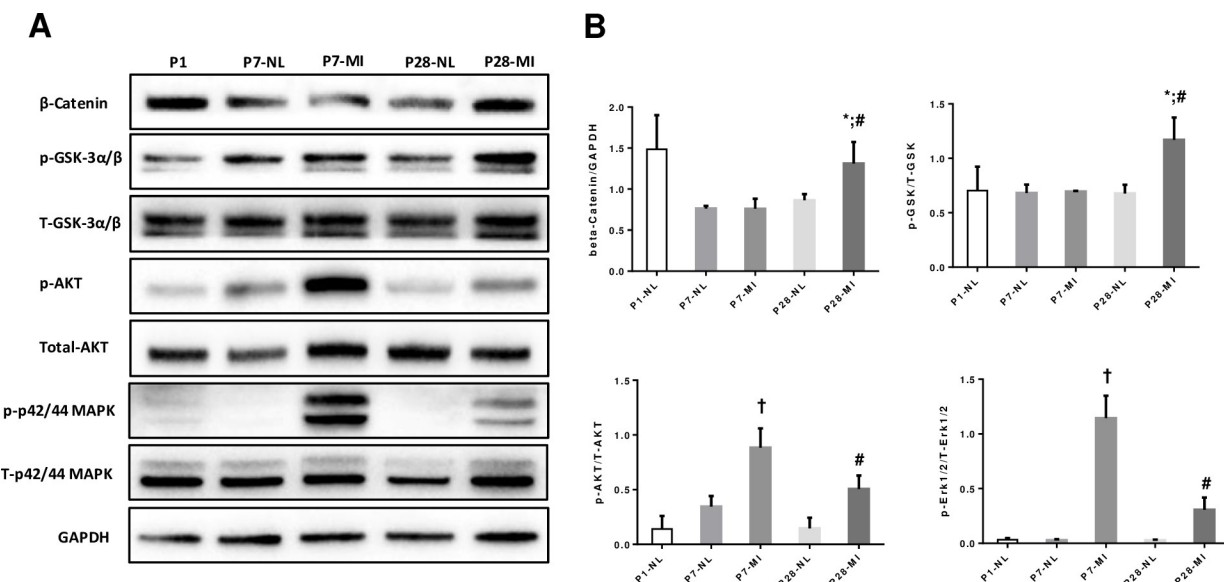

**Fig 6. Western blotting. LAD ligation at P1 activates key cell proliferation signaling pathways and downstream components.** The expression of β-catenin, phosphorylated (Pho-) GSK 3α/β, total GSK 3α/β, Pho-Akt, total Akt, Pho-p42/44 MAPK, and total p42/44 MAPK was evaluated via Western blot. GAPDH levels were also evaluated to confirm equal loading. **(A)** Representative Western blotting of hearts from each time point/group; **(B)** Compiled Wetern blotting data, n = 3 hearts each bar. T-test (2 tails) with Bonferroni correction. *, p<0.05 between P28-MI vs P28-NL; #, p < 0.05 vs P7-MI; †, p<0.05 vs P7-NL. Total protein blots were the same original blots stripped and reprobed.

recovery in zebrafish or mice [55, 59, 61, 64, 68–70], this study is among the first to establish their involvement in the regenerative response to cardiac injury in neonatal large mammals. We also classified the genes as early- or late-postnatal, depending on whether they were upregulated on P7 or P28, respectively, and several of the upregulated genes appeared to link two or more pathways. Notably, the expression of many genes that were upregulated on P7 declined to normal levels by P28, which suggests that they may be associated with early developmental processes that subside shortly after birth. Nevertheless, a number of genes (e.g., CSF1/CSF1R, MYC, JUN/JUND and ATF4) were upregulated at both time points, which suggested that at least some regenerative activity persisted in MI-P1 animals for up to 4 weeks after MI.

We have known that cardiac muscle regeneration can only occur in low vertebrate animals for some time. Recently, there are significant reports indicating that in small mammal (mouse) cardiomyocyte can regenerate [76, 77]. However, the myocyte proliferation capacity quickly lost before postnatal-day 7 [76, 77]. The observation of LV functional recovery in newborn humans by fixing the coronary perfusion immediately after birth [78], suggest that a significant level of cardiomyocyte proliferation can occur in large mammal during early neonatal age. We recently completed pilot studies examining the cardiac regenerative potential of neonatal hearts in large mammals [1, 2], and our results demonstrate that the neonatal porcine heart is capable of regenerating from MI for only the first 2 days of life. This regenerative capacity is mediated by the proliferation of pre-existing cardiomyocytes [2, 76, 79], and is lost before postnatal-day 7, when cardiomyocytes permanently exit the cell cycle. Using a pig model of MI at P1, the present study demonstrate a few signaling pathways that are associated with the, activation of key myocyte proliferation signaling pathways. Most importantly, as we begin to better understand the mechanisms that regulate the drastic early postnatal decline in cardiomyocyte proliferation, we may be able to manipulate these mechanisms to promote myocardial regeneration in adult, as well as pediatric, patients.

In conclusion, the studies presented here build upon the results from our previous report [19] by conducting pathway analysis to determine which signaling mechansims contribute to the regenerative capacity of newborn pig hearts. However, unlike conventional pathway analysis, we queried the KEGG Pathway Database first, to identify signaling cascades that are known to regulate the cell-cycle and proliferation, and then conducted differential gene expression analyses to confirm whether the identified pathways were upregulated in myocardial tissues from the hearts of pigs that underwent MI surgery one day after birth. Collectively, our findings demonstrate a comprehensive list of key regulators that controls cardiomyocyte exiting cell-cycle shortly after birth, which are significant informations for future in-vitro and in-vivo studies of myocardial regeneration in large mammals.

## Supporting information

**S1 Raw images.**
(PDF)

**S1 File.**
(ZIP)

**S1 Data.**
(XLSX)

## Author Contributions

**Conceptualization:** Eric Zhang, Jake Y. Chen.

**Data curation:** Eric Zhang.

**Formal analysis:** Eric Zhang, Thanh Nguyen, Son Do Hai Dang, Jake Y. Chen.

**Funding acquisition:** Eric Zhang.

**Investigation:** Eric Zhang, Weihua Bian, Gregory P. Walcott.

**Methodology:** Eric Zhang, Thanh Nguyen.

**Project administration:** Gregory P. Walcott.

**Software:** Eric Zhang, Son Do Hai Dang.

**Supervision:** Eric Zhang.

**Validation:** Eric Zhang, Meng Zhao.

**Visualization:** Eric Zhang, Thanh Nguyen, Son Do Hai Dang.

**Writing – original draft:** Eric Zhang, Thanh Nguyen, Jake Y. Chen.

**Writing – review & editing:** Eric Zhang, Thanh Nguyen, Meng Zhao, Jake Y. Chen.

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
