## [Decision Letter · Decision Letter 0]

6 Jan 2020

PONE-D-19-33077

Identifying the Key Regulators that Promote Cell-cycle Activity in the Hearts of Early Neonatal Pigs after Myocardial Injury

PLOS ONE

Dear Dr. Zhang,

Thank you for submitting your manuscript to PLOS ONE. After careful consideration, we feel that it has merit but does not fully meet PLOS ONE’s publication criteria as it currently stands. Therefore, we invite you to submit a revised version of the manuscript that addresses the points raised during the review process.

We would appreciate receiving your revised manuscript by Feb 20 2020 11:59PM. To enhance the reproducibility of your results, we recommend that if applicable you deposit your laboratory protocols in protocols.io, where a protocol can be assigned its own identifier (DOI) such that it can be cited independently in the future. For instructions see: http://journals.plos.org/plosone/s/submission-guidelines#loc-laboratory-protocols

We look forward to receiving your revised manuscript.

Kind regards,

Guo-Chang Fan, PhD

Academic Editor

PLOS ONE

Journal Requirements:

3. We note that you are reporting an analysis of a microarray, next-generation sequencing, or deep sequencing data set. PLOS requires that authors comply with field-specific standards for preparation, recording, and deposition of data in repositories appropriate to their field. Please upload these data to a stable, public repository (such as ArrayExpress, Gene Expression Omnibus (GEO), DNA Data Bank of Japan (DDBJ), NCBI GenBank, NCBI Sequence Read Archive, or EMBL Nucleotide Sequence Database (ENA)). In your revised cover letter, please provide the relevant accession numbers that may be used to access these data. For a full list of recommended repositories, see http://journals.plos.org/plosone/s/data-availability#loc-omics or http://journals.plos.org/plosone/s/data-availability#loc-sequencing.

"No any financial conflict interests"

Please provide an amended Funding Statement that declares *all* the funding or sources of support received during this specific study (whether external or internal to your organization) as detailed online in our guide for authors at http://journals.plos.org/plosone/s/submit-now.  

Please state what role the funders took in the study.  If any authors received a salary from any of your funders, please state which authors and which funder. If the funders had no role, please state: "The funders had no role in study design, data collection and analysis, decision to publish, or preparation of the manuscript."

Reviewers' comments:

Reviewer's Responses to Questions

**Comments to the Author**

1. Is the manuscript technically sound, and do the data support the conclusions?

Reviewer #1: Yes

Reviewer #2: Yes

Reviewer #3: Partly

2. Has the statistical analysis been performed appropriately and rigorously? 

Reviewer #1: Yes

Reviewer #2: Yes

Reviewer #3: Yes

3. Have the authors made all data underlying the findings in their manuscript fully available?

Reviewer #1: Yes

Reviewer #2: Yes

Reviewer #3: Yes

4. Is the manuscript presented in an intelligible fashion and written in standard English?

Reviewer #1: Yes

Reviewer #2: Yes

Reviewer #3: Yes

5. Review Comments to the Author

Reviewer #1: The mammal hearts have been thought to be a post-mitotic tissue for decades till recent studies demonstrated that neonatal rodent and porcine hearts can regenerate following injuries. However, the regenerative capacity of the hearts losses when the animal pass p7 after birth. Thus, to determine the cellular and molecular mechanisms of mammal neonatal heart regeneration is critical for regenerative medicine. In the current study, the authors performed RNA sequence analysis to determine the key signaling pathways that activate after myocardial infarction injury. They found that five signaling pathways (MAPK, Hippo, cAMP, JAK/STAT, and Ras) were upregulated following neonatal cardiac injury, indicates they might contribute to regenerative capacity of neonatal porcine hearts. In my personal opinion, the current study could open several new research directions for studying cardiac regeneration. Thus, it is an important study in this research field. I only have some minor comments:

1. The authors should add statistical analysis results in their figures and method section.

2. In the discussion section, the authors might want to add some comments on potential approaches to regulate these important pathways to enhance adult cardiac regeneration.

3. The resolution of the figures need to be enhanced for publication.

Reviewer #2: This is a clear and well-written manuscript. The introduction is relevant and

theory based. The methods are appropriate. There are only two issues associated with publication:

1) what is the reason of selecting pig for experiment among other mammals?

2) How would you explain the novelty of your work in compare with "MicroRNA expression, targeting, release dynamics and early-warning biomarkers in acute cardiotoxicity induced by triptolide in rats" Biomed Pharmacother 2019 Mar;111:1467-1477?

Reviewer #3: In this manuscript, Zhang et al. performed RNA-seq on the neonatal pigs after myocardial injury to identify key regulators promoting cell-cycle activity. However, this manuscript is limited to pathway analysis without further validation or mechanism studies. The detailed points are as follows:

1 All these five pathways shown in this manuscript should be examined further at least at protein level, such as IF or western blot.

2 The RNA-seq is perform in the heart tissue with mixture of difference cell types. The pathway identified by this approach is very likely not cardiomyocyte-specific. For example, the up-regulation of CSF and CSF1R could be from fibroblasts and monocytes respectively. Please make these five signalings more specific by using IF to probe the protein of interest.

3 The unit of y axis in Fig.1B, 2B and 4B are not clearly labelled.

6. PLOS authors have the option to publish the peer review history of their article (what does this mean?). If published, this will include your full peer review and any attached files.

Reviewer #1: No

Reviewer #2: No

Reviewer #3: No

---

## [Author Response · Author response to Decision Letter 0]

12 Mar 2020

Response to Reviewers

Manuscript # PONE-D-19-33077

“Identifying the Key Regulators that Promote Cell-cycle Activity in the Hearts of Early Neonatal Pigs after Myocardial Injury”

By Eric Y Zhang et al

We appreciate the editor and reviewers’ careful and constructive critique of our manuscript. A point-by-point response to their comments are provided as follows.

Reviewer #1: The mammal hearts have been thought to be a post-mitotic tissue for decades till recent studies demonstrated that neonatal rodent and porcine hearts can regenerate following injuries. However, the regenerative capacity of the hearts losses when the animal pass p7 after birth. Thus, to determine the cellular and molecular mechanisms of mammal neonatal heart regeneration is critical for regenerative medicine. In the current study, the authors performed RNA sequence analysis to determine the key signaling pathways that activate after myocardial infarction injury. They found that five signaling pathways (MAPK, Hippo, cAMP, JAK/STAT, and Ras) were upregulated following neonatal cardiac injury, indicates they might contribute to regenerative capacity of neonatal porcine hearts. In my personal opinion, the current study could open several new research directions for studying cardiac regeneration. Thus, it is an important study in this research field. I only have some minor comments:

1. The authors should add statistical analysis results in their figures and method section.

Response: Thank you. It is done accordingly in the revised manuscript on page 6 the 2nd paragraph of the revised manuscript as follows:

“Sequencing data were processed in the Cheaha computer cluster at UAB (https://docs.uabgrid.uab.edu/wiki/cheaha). Paired-end fastq files were evaluated with bthe trim-galore [1] package for quality assessments of the input sequencing files, and mapping was conducted with the Ensembl pig genome assembly (Sscrofa11.1) [2] and STAR package v2.5.2 [3] to produce the sorted BAM (Binary Alignment Map) file format; then, the sorted BAM file was converted into the SAM (Sequence Alignment Map) format with SAMtools [4], and mapped transcripts (raw expression) were counted with the HTSeq/0.6.1 package [5]. Expression levels were normalized and differential gene expressions (log2 of the fold-change and p-value) were analyzed with Deseq2 software [6]. In 3x3 comparsions, the probability of observing as extreme and more than ‘all A-samples strictly greater than (or less than) B-samples’ would be 0.59 = 0.002. This is the p-value for the statistical result above. The p-values other cases are computed as in [51]”

2. In the discussion section, the authors might want to add some comments on potential approaches to regulate these important pathways to enhance adult cardiac regeneration.

Response: Thank you for this important point. The following paragraph has been added on page 9, the last paragraph to page 10 the end of first paragraph, of the revised manuscript:

“We have known that cardiac muscle regeneration can only occur in low vertebrate animals for osmetime. Recently, there are significant reports indicating that in small mammal (mouse) cardiomyocyte can regenerate [9, 10]. However, the myocyte proliferation capacity quickly lost before postnatal-day 7 [9, 10]. The observation of LV functional recovery in newborn humans by fixing the coronary perfusion immediately after the birth [11], suggest that a significant level of cardiomyocyte proliferation can occur in large mammal during early neonatal age. We recently completed a pilot study examining the cardiac regenerative potential of neonatal hearts in large mammals [7, 8], and our results demonstrate that the neonatal porcine heart is capable of regenerating from MI for only the first 2 days of life. This regenerative capacity is mediated by the proliferation of pre-existing cardiomyocytes [7-9], and is lost before postnatal-day 7, when cardiomyocytes permanently exit the cell cycle. Using a pig model of MI at P1, the present study demonstrate a few signaling pathways that are associated with the, activation of key myocyte proliferation signaling pathways. Most importantly, as we begin to better understand the mechanisms that regulate the drastic early postnatal decline in cardiomyocyte proliferation, we may be able to manipulate these mechanisms to promote myocardial regeneration in adult, as well as pediatric, patients.”

3. The resolution of the figures need to be enhanced for publication.

Response: Yes, we agree. The resolution of all the figures have been enhanced accordingly 

 

Reviewer #2: This is a clear and well-written manuscript. The introduction is relevant and theory based. The methods are appropriate. There are only two issues associated with publication:

1) what is the reason of selecting pig for experiment among other mammals?

Response: Thank you for asking this important question. For a long time, we have known that cardiac muscle regeneration can only occur in low vertebrate animals. Recently, there are significant reports that small mammal (mouse) cardiomyocyte can regenerate [9, 10]. This regenerative capacity is mediated via the proliferation of pre-existing cardiomyocytes [9, 10]. However, the myocyte proliferation capacity quickly lost before postnatal-day 7 [9, 10]. The observation of LV functional recovery in newborn humans by fixing the coronary perfusion immediately after the birth [11], suggest that a significant level of cardiomyocyte proliferation can occur in large mammal during early neonatal age. 

The following paragraph has been added on page 9, the last paragraph to page 10 the end of first paragraph, of the revised manuscript:

“We have known that cardiac muscle regeneration can only occur in low vertebrate animals for sometime. Recently, there are significant reports indicating that in small mammal (mouse) cardiomyocyte can regenerate [9, 10]. However, the myocyte proliferation capacity quickly lost before postnatal-day 7 [9, 10]. The observation of LV functional recovery in newborn humans by fixing the coronary perfusion immediately after the birth [11], suggest that a significant level of cardiomyocyte proliferation can occur in large mammal during early neonatal age. We recently completed a pilot study examining the cardiac regenerative potential of neonatal hearts in large mammals [7, 8], and our results demonstrate that the neonatal porcine heart is capable of regenerating from MI for only the first 2 days of life. This regenerative capacity is mediated by the proliferation of pre-existing cardiomyocytes [7-9], and is lost before postnatal-day 7, when cardiomyocytes permanently exit the cell cycle. Using a pig model of MI at P1, the present study demonstrate a few signaling pathways that are associated with the, activation of key myocyte proliferation signaling pathways. Most importantly, as we begin to better understand the mechanisms that regulate the drastic early postnatal decline in cardiomyocyte proliferation, we may be able to manipulate these mechanisms to promote myocardial regeneration in adult, as well as pediatric, patients.”

2) How would you explain the novelty of your work in compare with "MicroRNA expression, targeting, release dynamics and early-warning biomarkers in acute cardiotoxicity induced by triptolide in rats" Biomed Pharmacother 2019 Mar;111:1467-1477?

Response: Using a large mammal model, the current study seeks to examine the mechanisms that regulate the drastic early postnatal decline in cardiomyocyte proliferation. And in future, we aim to manipulate these mechanisms to promote myocardial regeneration in adult. Using a mall mammal model, the aforementioned study seeks to examine the mechanisms of acute cardiotoxicity induced by triptolid, in terms of microRNA expression, targeting, release dynamics as early biomarkers. 

 

Reviewer #3: In this manuscript, Zhang et al. performed RNA-seq on the neonatal pigs after myocardial injury to identify key regulators promoting cell-cycle activity. However, this manuscript is limited to pathway analysis without further validation or mechanism studies. 

The detailed points are as follows:

1 All these five pathways shown in this manuscript should be examined further at least at protein level, such as IF or western blot. 

Response: We agree and have conducted Western blot analyses of the expression of MAPK1/3, which mediates MAPK signaling; Akt, which mediates the cell-cycle regulatory activity of the JAK/STAT, cAMP, and RAS pathways; and the Wnt regulators GSK 3α/β and β-catenin, which act on downstream factors of the Hippo pathway. The amount of activated (i.e., phosphorylated) 42/44-MAPK and Akt in MI P1 apex resection (AR) hearts at P7 and P28 was significantly greater in AR hearts than in age matched normal hearts at each time points; thus, MAPK, Wnt/β-catenin and Akt signaling were upregulated for at least 28 days after AR induction at P1. 

This figure and following paragraph have been included on page 6, paragraph 3 , and Figure 6, of the revised manuscript. 

Western blotting. Tissues were lysed in Mammalian Protein Extraction Reagent (Fisher scientific, PI78501) with protease (Sigma, 04693116001) and phosphatase (Sigma, P0044) inhibitors; then, the lysates were denatured at 100 °C for 6 min, separated in a 4-20% precast gel (Bio-rad, 4568093), and transferred onto a PVDF membrane (Bio-rad, 1704156). The membrane was incubated with 5% non-fat milk (Bio-rad, 1706404) for 30 min, with primary antibodies at 4 °C overnight, and then with horseradish-peroxidase (HRP)–conjugated secondary antibodies for 30 min. Protein bands were detected with the chemiluminescent HRP substrate (Millipore, WBKLS0500) in a ChemiDocTM Imaging System (Bio-rad). Phosphorylated proteins were detected first; then, the blots were treated with stripping buffer (Fisher scientific, PI21059) before total protein expression was evaluated.

2 The RNA-seq is perform in the heart tissue with mixture of difference cell types. The pathway identified by this approach is very likely not cardiomyocyte-specific. For example, the up-regulation of CSF and CSF1R could be from fibroblasts and monocytes respectively. Please make these five signaling more specific by using IF to probe the protein of interest.

Response: The present investigation 

3 The unit of y axis in Fig.1B, 2B and 4B are not clearly labelled.

Response: Thank you . The unit of y axis in Fig.1B, 2B and 4B are clearly labelled in the revised manuscript. 

 

REFRENCES 

1. Zhou H, Dickson ME, Kim MS, Bassel-Duby R, Olson EN. Akt1/protein kinase B enhances transcriptional reprogramming of fibroblasts to functional cardiomyocytes. Proc Natl Acad Sci U S A. 2015;112(38):11864-9. Epub 2015/09/12. doi: 10.1073/pnas.1516237112. PubMed PMID: 26354121; PubMed Central PMCID: PMCPMC4586885.

2. Cunningham F, Achuthan P, Akanni W, Allen J, Amode MR, Armean IM, et al. Ensembl 2019. Nucleic Acids Res. 2019;47(D1):D745-D51. Epub 2018/11/09. doi: 10.1093/nar/gky1113. PubMed PMID: 30407521; PubMed Central PMCID: PMCPMC6323964.

3. Dobin A, Davis CA, Schlesinger F, Drenkow J, Zaleski C, Jha S, et al. STAR: ultrafast universal RNA-seq aligner. Bioinformatics. 2013;29(1):15-21. Epub 2012/10/30. doi: 10.1093/bioinformatics/bts635. PubMed PMID: 23104886; PubMed Central PMCID: PMCPMC3530905.

4. Li H, Handsaker B, Wysoker A, Fennell T, Ruan J, Homer N, et al. The Sequence Alignment/Map format and SAMtools. Bioinformatics. 2009;25(16):2078-9. Epub 2009/06/10. doi: 10.1093/bioinformatics/btp352. PubMed PMID: 19505943; PubMed Central PMCID: PMCPMC2723002.

5. Anders S, Pyl PT, Huber W. HTSeq--a Python framework to work with high-throughput sequencing data. Bioinformatics. 2015;31(2):166-9. Epub 2014/09/28. doi: 10.1093/bioinformatics/btu638. PubMed PMID: 25260700; PubMed Central PMCID: PMCPMC4287950.

6. Love MI, Huber W, Anders S. Moderated estimation of fold change and dispersion for RNA-seq data with DESeq2. Genome Biol. 2014;15(12):550. Epub 2014/12/18. doi: 10.1186/s13059-014-0550-8. PubMed PMID: 25516281; PubMed Central PMCID: PMCPMC4302049.

7. Zhu W, Zhang E, Zhao M, Chong Z, Fan C, Tang Y, et al. Regenerative Potential of Neonatal Porcine Hearts. Circulation. 2018. Epub 2018/07/22. doi: 10.1161/CIRCULATIONAHA.118.034886. PubMed PMID: 30030418.

8. Ye L, D'Agostino G, Loo SJ, Wang CX, Su LP, Tan SH, et al. Early Regenerative Capacity in the Porcine Heart. Circulation. 2018. doi: 10.1161/CIRCULATIONAHA.117.031542. PubMed PMID: 30030417.

9. Porrello ER, Mahmoud AI, Simpson E, Hill JA, Richardson JA, Olson EN, et al. Transient regenerative potential of the neonatal mouse heart. Science. 2011;331(6020):1078-80. doi: 10.1126/science.1200708. PubMed PMID: 21350179; PubMed Central PMCID: PMCPMC3099478.

10. Porrello ER, Mahmoud AI, Simpson E, Johnson BA, Grinsfelder D, Canseco D, et al. Regulation of neonatal and adult mammalian heart regeneration by the miR-15 family. Proceedings of the National Academy of Sciences of the United States of America. 2013;110(1):187-92. doi: 10.1073/pnas.1208863110. PubMed PMID: 23248315; PubMed Central PMCID: PMCPMC3538265.

11. Haubner BJ, Schneider J, Schweigmann U, Schuetz T, Dichtl W, Velik-Salchner C, et al. Functional Recovery of a Human Neonatal Heart After Severe Myocardial Infarction. Circ Res. 2016;118(2):216-21. doi: 10.1161/CIRCRESAHA.115.307017. PubMed PMID: 26659640.

---

## [Editor Report · Decision Letter 1]

27 Apr 2020

Identifying the Key Regulators that Promote Cell-cycle Activity in the Hearts of Early Neonatal Pigs after Myocardial Injury

PONE-D-19-33077R1

Dear Dr. Zhang,

We are pleased to inform you that your manuscript has been judged scientifically suitable for publication and will be formally accepted for publication once it complies with all outstanding technical requirements.

With kind regards,

Guo-Chang Fan, PhD

Academic Editor

PLOS ONE

Additional Editor Comments (optional):

Revised manuscript has fully addressed concerns raised by previous reiewers, resulting in an acceptable version.
---

## [Editor Report · Acceptance letter]

17 Jul 2020

PONE-D-19-33077R1 

Identifying the Key Regulators that Promote Cell-cycle Activity in the Hearts of Early Neonatal Pigs after Myocardial Injury 

Dear Dr. Walcott:

I'm pleased to inform you that your manuscript has been deemed suitable for publication in PLOS ONE. Congratulations! Your manuscript is now with our production department. 

Kind regards, 

on behalf of

Dr. Guo-Chang Fan 

Academic Editor

PLOS ONE